# Vaxtherapy, a Multiphase Therapeutic Protocol Approach for Longvax, the COVID-19 Vaccine-Induced Disease: Spike Persistence as the Core Culprit and Its Downstream Effects

**DOI:** 10.3390/diseases13070204

**Published:** 2025-06-30

**Authors:** Jose Crespo-Barrios

**Affiliations:** Department of Proteomics, Regeneratics, 28019 Madrid, Spain; jose.crespobarrios@regeneratics.com

**Keywords:** SARS-CoV-2, COVID-19, vaccine injured, vaccine-induced disease, mRNA, therapy, protocol

## Abstract

**Background/Objectives:** Chronic illness after COVID-19 vaccination (longvax) lacks a therapeutic protocol anchored in pathophysiology. Persistent vaccine derived spike protein appears to trigger microvascular fibrin amyloid microclots, immune dysfunction, pathogen reactivation and multisystem injury. This article proposes an integrative approach, Vaxtherapy, to tackle these mechanisms. **Methods:** A narrative synthesis of peer reviewed literature from 2021 to 2025 on spike related injury and vaccine adverse events was conducted, supplemented by clinical case series and mechanistic observations from long COVID. The findings were arranged into a four stage therapeutic sequence ordered by pathophysiological precedence. **Results:** Stage one aims to reopen hypoperfused tissue through oral fibrinolytics that degrade fibrin amyloid resistant microclots; stage two intends to neutralise circulating or tissue bound spike via a receptor binding domain monoclonal antibody cocktail; stage three seeks to eliminate reactivated viral or microbial reservoirs with targeted antivirals or antimicrobials once perfusion is improved; and stage four aspires to support tissue repair with mitochondrial supplements and, when indicated, cell based therapies. Omitting or reordering stages may reduce efficacy or foster resistance. **Conclusions:** This hypothesis driven framework outlines a biologically plausible roadmap for longvax research. By matching interventions to specific mechanisms (fibrinolysis, spike neutralisation, pathogen clearance and regeneration), it aims to guide controlled trials and compassionate pilot programs directed at durable recovery rather than chronic symptom management.

## 1. Introduction

COVID-19 vaccine injury became evident once it was observed that the spike protein alone could account for a substantial portion of the multisystemic damage attributed to the virus SARS-CoV-2 [1,2,3]. This became particularly clear when it was determined that the intravascular access of the spike protein was the critical factor [2]. Early studies demonstrating this phenomenon are therefore of particular relevance. The author produced an instructive video explaining this process [4]. Years later, this has been established as the standard pathophysiological explanation of the phenomenon [5,6], although there were also some advanced teams on that regard [7].

A detailed description of the multitude of injuries caused by this protein is not provided here, both because it has already been comprehensively covered in the literature [8] and because it falls outside the scope of this article. It should nonetheless be underscored in broad terms that the spike protein can produce multiorgan damage [1,9], exert effects at the cellular level (including nuclear alterations) [10], disrupt cellular crosstalk, modulate immune functions involving white blood cells [11], impair cellular respiration in red blood cells [12], and induce syncytial formation [13], thereby encouraging viral reservoirs [8,14,15]. Moreover, it affects the microvasculature and can lead to coagulopathies [2,16], which foster hypoperfusion. This, in turn, contributes to nutritional, communicative, and immunological deficits, promotes pathogen reactivation, potentially drives tumorigenesis or carcinogenesis, and triggers other downstream consequences; see Figure 1.

Evidence for spike persistence has been accumulating since 2021. S1 fragments have been detected inside CD16^+^ monocytes for up to 15 months, sustaining a low-grade inflammatory milieu that mirrors post-acute COVID-19 sequelae [9,17]. Molecular mimicry between conserved spike epitopes and host G-protein-coupled receptors can elicit persistent functional autoantibodies that perpetuate tissue injury and dysautonomia [18]. In parallel, spike-induced endothelial damage promotes microangiopathy and coagulopathies [2,16], leading to regional hypoperfusion. The resulting nutrient, signaling, and immune deficits favor latent-pathogen reactivation, may fuel oncogenic processes, and generate a spectrum of downstream complications that demand multidisciplinary care [19]; see Figure 1.

Beyond the interest of affected individuals in having their illness recognized and obtaining social compensation, there lies a more fundamental concern: how they can regain their lives. Patients do not wish to be supported indefinitely; rather, they want to restore their own health. Consequently, it is of paramount importance to begin discussing serious therapeutic options that go beyond merely managing downstream symptoms. Specifically, such options should enable patients to recover their health without relying on ongoing pharmacological treatments that may entail direct financial conflicts of interest in the pharmaceutical sector. The ultimate aim, therefore, is to substantially reduce pharmacological dependence and significantly improve patients’ quality of life.

## 2. Common and Disparate Characteristics Between Longvax and Long COVID

### 2.1. Clinical: Symptoms, Pathophysiology, and Biomarkers

#### 2.1.1. Shared Background

Although most chronic manifestations that follow either SARS-CoV-2 infection or vaccination overlap (largely because of spike persistence and its downstream inflammatory cascade), the literature documents clinically and biologically actionable differences that allow the two entities to be separated in practice.

#### 2.1.2. Persistent Ipsilateral Lymphadenopathy

Longvax is typified by prolonged ipsilateral axillary lymph node enlargement: imaging series after mRNA or adenoviral vaccination report prevalence rates approaching 33%, with a mean ultrasound resolution time near 102 days and pharmacovigilance cases lasting more than 6 months (median 230 days) [20,21,22]. In contrast, lymphadenopathy after infection is anecdotal: a PET-CT case described persistent cervical, thoracic, and abdominal nodes four months after mild COVID-19 [23], and a three-patient lupus/MCTD (Mixed Connective Tissue Disease) series noted diffuse cervical axillary nodes 4 weeks post-infection [24]. No prospective cohort has linked unilateral axillary nodes to long COVID.

#### 2.1.3. Vaccine-Induced Immune Thrombotic Thrombocytopenia (VITT)

Among adenoviral vector recipients, the hallmark complication is VITT thrombosis at unusual sites plus platelets < $50\times 10^{9}\;L^{-1}$, extreme D-dimer within 30 days, and platelet-activating anti-PF4 IgG [25,26]. PF4, a strongly cationic chemokine, binds highly anionic hexon capsomers and free DNA within ChAdOx1/Ad26 virions [27], creating neo-epitopes that drive high-affinity anti-PF4 IgG production within days [28]. Spike can bind PF4 in vitro but fails to form the conformation that elicits platelet-activating antibodies; VITT anti-PF4 does not cross-react with spike [29]. Accordingly, >1 billion mRNA doses have yielded only sporadic PF4-dependent events [30]. Long COVID, while prothrombotic through endothelial dysfunction and microclots, lacks the triad of profound thrombocytopenia, sky-high D-dimer, and activating anti-PF4 [31,32].

#### 2.1.4. Nucleocapsid and Membrane Protein-Driven Effects Contribute to Differentiate Long COVID from Longvax

Certain presentations point to a pathology specific to N and M proteins rather than spike: the N protein engagement of MASP-2 (Mannan-Binding Lectin-Associated Serine Protease 2) triggers complement-mediated purpuric vasculitis [33]; N protein-driven NLRP3 hyperactivation yields fulminant IL-1β storms [34]; and M-mediated TBK1 degradation silences type-I interferon, producing high viral loads with silent hypoxemia [35]. Spike can prime IL-1β release via TLR2/4, yet this response is weaker, disappears with TLR blockade, and is absent in macrophages exposed to fully glycosylated, endotoxin-free spike [36,37,38]; a robust, TLR-independent IL-1β surge therefore argues for N/M proteins involvement.

#### 2.1.5. Differential Biomarker Panel

In VITT, D-dimer routinely exceeds 10 µg mL^−1^ within 30 days, and activating anti-PF4 IgG are virtually pathognomonic [29,39]. Beyond serology (anti-S^+^/anti-N^−^/anti-M^−^), non-coding RNA profiling adds resolution: an extracellular vesicle miRNA triad (miR-223-3p, miR-24-3p, and miR-15b-5p) persists at 3–6 months after mRNA vaccination but is absent in long COVID [40]; conversely, miR-200c-3p, miR-766-3p, and miR-142-3p fall and extravesicular miR-34a rises only after infection [41,42]. The lncRNA NEAT1 is likewise upregulated in post-infectious cohorts and baseline in longvax [43].

More experimentally, there is incipient but growing evidence of the continuance of the modified vaccine mRNA instructions that could shed light to a line of future biomarkers [44,45,46].

Combining these RNA markers with anti-S/anti-N serology yields a high-specificity algorithm for difficult cases.

## 3. Description of the Model for the Vaxtherapy Protocol

### 3.1. First Step: Ameliorating Microvascular Hypoperfusion with Fibrinolytic Agents

One of the defining characteristics of 2021 and 2022 longvax cases, which they share with individuals experiencing long COVID after being infected prior to the Omicron variant, is the impact of the corresponding spike variants. The evolution from pre-Omicron to Omicron spike is a key factor that led to a drastic reduction in COVID-19 severity, explained by the pre-Omicron capacity to inhibit fibrinolysis of the resultant coagulopathy [47]. This had two major consequences. First, when vaccinated individuals presented to the emergency room, they did not necessarily show elevated D-dimer levels (a marker of coagulopathy) precisely because it is a by-product of the fibrinolytic process, which did not occur as readily as in the post-Omicron period. As a result, patients did not receive the appropriate treatments; had they received those therapies in time, it is possible that longvax sequelae and related complications might not have developed or would have been significantly less severe.

In other words, persistent microclots (referred to here as “fibrin amyloid-resistant microclots” or FARMs) are formed and resist being eliminated. These clots block defensive cells, nutrients, and essential signaling from reaching affected areas, leading to severe consequences [8,48]. As we detailed in our 2024 study, this difference between the pre-Omicron and post-Omicron spike is critical [8]. Logically, if these FARMs hinder defensive agents from accessing hypoperfused regions, therapies such as polyclonal antibodies (and indeed any other treatments) cannot fully exert their effects. Hence, the active use of fibrinolytic agents constitutes the first line of attack in the therapeutic protocol for longvax. This fact and the causes behind it are outlined in our previous work on long COVID cure strategies [8], and later reflected in the work of Polybio [49]. For similar reasons, other therapeutic approaches within the protocol must be applied in the correct order of precedence; see Figure 2. In the classification from the 2024 paper (and originally from 2022), there are two types of longvax: the one marked by spike persistence (“LC-B”, according to [8]), and the one marked by the hypothesis of a chronic generation of spikes via persistent mRNA instructions (“LC-C2” according to [8]), based on growing evidence [44,45], although more research is required. The global context that encompasses the classification and explanation from LC-A to LC-D and the inclusion of the mentioned LC-B and LC-C2 is depicted in Figure 3. Likewise, it is of limited benefit to use agents that enhance cellular respiration (e.g., ubiquinol, PQQ), nutrients, or broad-spectrum defensive measures if they cannot reach their target tissues.

Therefore, the fibrinolytic agents recommended in that paper (such as nattokinase, serrapeptase, lumbrokinase, or others including papain, bromelain, etc.) should be considered first-line therapy for longvax when addressing therapeutic studies. It is important to emphasize that, barring specific patient-related exceptions, enhancing fibrinolysis is generally preferred over anticoagulation, because systemic vascular permeability could worsen the clinical picture (e.g., increased risk of hemorrhage, possible microsepsis via translocation from the damaged intestinal tract, etc.). At the same time, one must consider drug interactions (for instance, patients on chronic antiplatelet therapy prior to the longvax curative protocol). Finally, it remains necessary to monitor coagulation markers when applying fibrinolytics, since their effects are interdependent.

### 3.2. Second Step:Neutralization of the Spikes with Multimodal Monoclonal Approach

The next step is to deactivate the primary factor contributing to the issue described above (aside from the persistence of spike-producing instructions), namely the persistence of spike proteins (LC-B). The receptor-binding domain (RBD) is especially pertinent as a therapeutic target due to its high functional stress, which limits its mutational capacity, as well as its degree of exposure. This is particularly significant in post-vaccination cases, since unlike long COVID cases that involve persistent viral replication (LC-C1 and beyond), the spectrum of potential spike mutations in post-vaccination LC-B might be notably narrower [8]. This fact is highlighted in Figure 4, panel A. In practical terms, if no prior infection or other influencing factors are present, individuals affected by the SARS-CoV-2 vaccine typically display only a single spike variant. Consequently, neutralizing these spike proteins with a cocktail of monoclonals is especially advantageous.

In particular, sotrovimab, casirivimab, and imdevimab are capable of co-participation; see Figure 4 [50,51,52]. Once they bind, the spike protein is neutralized, ceasing to induce the hyperactivation of the immune system and thereby alleviating immune exhaustion. This also protects against the direct damage caused by the spike (e.g., inhibition of immune synapses), which, on its own, allows for other functions to proceed, such as the control of pathogenic reservoirs (not limited to viruses) and the reestablishment of constructive immune processes (e.g., switching macrophages to the M2 phenotype).

### 3.3. Third Step: Pathogen Clearance

Once the walls have been demolished (i.e., FARMs), as well as their constructors (i.e., active functional spikes) the next step is to promote recovery from reactivated pathogens. After performing a thorough assessment to identify which pathogens have truly become active, targeted measures are taken against them. Antiviral options (e.g., for Epstein–Barr virus, varicella zoster virus, among others) or antibiotics constitute the subsequent tier of recovery. Although less common, antiparasitic treatments [53] could also be considered at this stage. Note that the sequence in which these interventions are applied remains crucial at this stage, since there is little benefit in proceeding with this protocol phase if the earlier ones have not been adequately resolved; in fact, doing so could be counterproductive, potentially leading to the development of resistance. In cases where patients do not tolerate certain medications well, it may be advisable to reinforce or prioritize regenerative strategies so that drug metabolism is sufficiently restored to facilitate this treatment phase.

Finally, as discussed in a previous publication [8], both drug tolerance, half-life, as well as the capacity of these treatments to penetrate tissues, are important factors (one of the many parallels between the original 2024 article [8] and the likely subsequent 2025 Polybio paper [49]). Completing this step in the protocol not only halts ongoing damage and eases the immune system’s workload but also substantially enhances the body’s regenerative capacity by restoring intercellular communication and intracellular proteodynamics. Furthermore, relieving the immune system of these distractions supports an even greater degree of recovery.

### 3.4. Fourth Step: Supplementation and Regenerative

Once the above steps have been completed (and aside from any tolerogenic considerations related to the medications used in the previous phase) the focus now shifts to repairing the damage. This includes restoring affected cell populations (e.g., the immune system, intermediate intratissue stem cells) as well as structural cells (e.g., myocardial tissue, vasculature). Completing these measures effectively addresses the “LC-D” category according to the previous classification in Figure 3, that is, damage deemed structural or irreparable without regenerative medical intervention.

With respect to supplementation (which has been the subject of extensive research) it now becomes genuinely effective for the aforementioned reasons, provided only the genuinely necessary supplements are used. Theoretically, if all prior steps have been successfully carried out, no supplementation at all would be needed. However, this is merely a theoretical upper bound; in practice, supervised supplementation under the guidance of knowledgeable professionals will always support recovery and regeneration in a safe manner. Substances highlighted in this context include ubiquinol, PQQ, and taurine for cellular respiration, along with antioxidant polyphenols (e.g., resveratrol). Other agents may be needed to correct specific deficiencies (such as vitamin B12 [54,55]) or other nutrients that appear statistically likely to be deficient.

## 4. Pathophysiological and Clinical Experience Background of the Protocol

### 4.1. Pathophysiology

The persistence of the spike protein and its downstream pathological effects have been well documented in both long COVID and vaccine-induced longvax syndromes [2,56,57,58]. These conditions share a common pathophysiological framework caused by the spike protein. It is primarily characterized by vascular and general hematological dysfunction, including immune system dysregulation, driven by prolonged exposure to the spike protein. Despite this understanding, no systematic, targeted therapeutic approach has been developed to address longvax.

We hypothesize that a structured, stepwise therapeutic strategy (one that prioritizes addressing the primary drivers of pathology first) will yield better clinical outcomes than symptomatic or non-sequential interventions. The core principle is that each phase of treatment facilitates the success of the next; failing to adhere to this sequence may not only reduce efficacy but could also worsen outcomes. Specifically, the proposed protocol is designed in four stages:

Restoration of tissue perfusion, i.e., securing access pathways; neutralization of spike protein, the primary cause of hypoperfusion and its downstream consequences; elimination of reservoirs, which prevent natural body regeneration and contribute to autoimmunity and other dysfunctions; and adjuvant therapies for regeneration and baseline recovery. This section outlines only the fundamental reasoning behind the chosen strategy, while the detailed protocol is described in the corresponding section.

#### 4.1.1. Perfusion Recovery as a Prerequisite for Therapeutic Efficacy

Consider, for example, the case of administering an antiviral for Epstein–Barr virus (EBV) in a longvax patient experiencing EBV reactivations, who also suffers from severe tissue hypoperfusion due to FARMs. As the name suggests (fibrin amyloid-like-resistant microclots), FARMs are highly persistent and hyperinflammatory microclots that obstruct necessary vascular access to tissues. This blockage impairs the delivery of essential circulating substances required for homeostasis, leading to hypoperfusion [59,60].

Given this pathophysiological understanding, it is reasonable to assume that the antiviral’s penetration into affected tissues will be significantly hindered if vascular access remains obstructed. Worse still, due to the reduced effective dose of the antiviral reaching the viral reservoirs, there is a risk that these reservoirs could develop resistance to the treatment. However, this issue is not exclusive to externally applied therapeutic agents; it also affects immune cell access, nutrient delivery, cellular communication, energy supply, and all other vital functions required for cellular defense and operation [61].

For this reason, following the same principle as resolutive strategies, securing open pathways must be ensured before applying treatments. In this context, FARMs represent the primary obstacle in circulation, and fibrinolytic agents are the first proposed solution to counteract them.

#### 4.1.2. Neutralization of the Spike Protein

Once intratissue access pathways have been secured, the next step is to neutralize the root spike protein that led to the formation of FARMs in the first place and its various downstream effects. To target the most functionally active and vulnerable region of the spike, a multi-monoclonal strategy is applied to the receptor-binding domain (RBD) [62], following the principles established in the previous 2024 study [8]. Deactivating the functional activity of spike proteins, among other benefits, helps maintain open vascular pathways during and after the therapeutic protocol.

#### 4.1.3. Elimination of Pathogenic Reservoirs

Viral reactivations are one of the primary concerns in both long COVID and longvax symptomatology. Once the barriers have been dismantled, it becomes possible to effectively target reservoirs with antivirals, antibiotics, or other agents, depending on the nature of the reservoir. Attempting this step before the previous ones would not only be ineffective but could also be counterproductive, increasing the risk of resistance to the applied treatments.

#### 4.1.4. Adjuvant Therapies for Regeneration and Recovery

Finally, once all factors that could reverse the regeneration process have been eliminated, this last step is implemented. Applying it prematurely would be illogical, as unresolved underlying factors could counteract the benefits, leading to unnecessary time delays, increased costs, and other complications. Therefore, it is imperative that this be the final step.

An exception could be made in cases where regeneration is specifically aimed at improving tolerance to the agents required for the earlier steps, or for other well-justified reasons.

### 4.2. Clinical Experience: Reproducibility

This protocol is based not only on the aforementioned pathophysiological foundations but also on clinical experience results for each step of the Vaxtherapy protocol.

#### 4.2.1. Clinical Experiences with Fibrinolytics

These agents already have clinical experience that supports the experimental initiation of the protocol. In particular, favorable data are available for nattokinase [63], serrapeptase [64], and lumbrokinase [65]. Although aspirin is primarily considered an antiplatelet agent, it also has fibrinolytic applications, as demonstrated in a randomized controlled trial (RCT) of 1000 individuals with suspected acute myocardial infarction [66].

There is also clinical experience with nattokinase, a serine proteinase from Bacillus subtilis. For instance, a clinical trial evaluated its effects on blood clotting and lipid levels associated with cardiovascular disease (CVD). The study included three groups: healthy volunteers, patients with cardiovascular risk factors, and dialysis patients. Participants took two capsules (each containing 2000 fibrinolysis units) daily for two months. Results showed a significant reduction in fibrinogen (9–10%), factor VII (7–14%), and factor VIII (17–19%), while blood lipid levels remained unchanged. No adverse effects were reported, suggesting nattokinase may support cardiovascular health [63].

Lumbrokinase, derived from Lumbricus rubellus, is a fibrinolytic enzyme that reduces fibrinogen (Fg) levels, aiding in ischemic stroke prevention. One key study found that it enhanced t-PA activity, decreased D-dimer levels, and improved blood viscosity without affecting coagulation, minimizing the risk of hemorrhage. After one year, carotid atherosclerosis markers improved, and the incidence of vascular events dropped by 4.7% [65].

Regarding serrapeptase, a proteolytic enzyme, there is also clinical evidence of its effectiveness. It exhibits strong fibrinolytic activity (1295 U/mg) and prevents blood coagulation at 150 U/mL. It achieves 96.6% clot lysis at 300 U/mL within 4 h at 37 °C. Optimal activity occurs at 37–40 °C and pH 9.0. Certain metal ions enhance its activity, while SDS and EDTA inhibit it. These findings suggest its potential as a health supplement or a therapeutic agent for thrombotic disease prevention [64].

#### 4.2.2. Clinical Experiences with Monoclonal Antibodies Targeting the Spike Protein

Sotrovimab is among the most suitable monoclonal antibodies (mAbs) for applications, based on clinical experience targeting the spike protein. Its selectivity index is significantly high for nearly all spike variants of interest, making it applicable to a broad range of vaccine-related spike proteins, including Omicron variants. This is particularly relevant since the entire protocol must initially be tested to obtain robust efficacy data, requiring a large sample population and, therefore, a broadly effective monoclonal antibody [67].

Clinical experience with sotrovimab includes trials such as COMET-ICE, a randomized, double-blind, placebo-controlled, phase III study that assessed its efficacy in non-hospitalized patients with mild to moderate COVID-19 at high risk for disease progression. The trial demonstrated a 79% reduction in hospitalization or death among participants receiving sotrovimab compared to placebo [68]. Another randomized clinical trial (RCT) compared the efficacy and adverse events of sotrovimab versus placebo in preventing disease progression. It found that sotrovimab significantly reduced the risk of hospitalization or death among high-risk patients by neutralizing the spike protein [69].

Regarding the combination of casirivimab and imdevimab, there is a case report involving a COVID-19 patient with humoral immunodeficiency, a complex scenario due to deficient antibody production. The study describes a 58-year-old woman treated with rituximab (an anti-CD20 monoclonal antibody) for follicular lymphoma, who experienced multiple COVID-19 relapses. She was successfully treated with the casirivimab/imdevimab cocktail, which led to viral clearance and a robust neutralizing antibody response. Her neutralizing antibody levels, which were undetectable before treatment, surged 55,700-fold two days post-administration and remained elevated for a month, suggesting prolonged passive immunity. A follow-up CT scan confirmed near-complete resolution of lung infiltrates. This case highlights casirivimab/imdevimab’s efficacy in B-cell-depleted patients who struggle to clear SARS-CoV-2, particularly when conventional treatments like corticosteroids and remdesivir fail [62].

In contrast with Fc-silenced formats, the dual monoclonal cocktail casirivimab + imdevimab (REGN-COV2) retains a wild-type IgG1 Fc domain and therefore exhibits measurable ADCC (Antibody-Dependent Cellular Cytotoxicity) and ADCP (Antibody-Dependent Cellular Phagocytosis) in vitro. Nevertheless, in >16,000 subjects enrolled in the pivotal trials, the rate of clinically relevant hypersensitivity was approximately 1% with no anaphylaxis reported, and regulators detected no evidence of antibody-dependent enhancement or Fc-driven immunopathology [70,71,72]

Post-marketing pharmacovigilance has documented only isolated severe acute reactions (<1/10,000) [73,74]. The current SmPC mitigates this residual risk by recommending a slow first infusion, pre-medication with antihistamine and/or corticosteroid when indicated, and observation for at least 60 min; under these precautions the combination has shown an acceptable safety profile even with monthly repeat dosing. Where Fc-dependent effector activity is undesirable, purely neutralizing options are available: antibodies expressed on IgG1 or IgG4 backbones bearing the TM (L234F/L235E/P331S) ± YTE (M252Y/S254T/T256E) mutations (exemplified by AZD7442) abrogate binding to Fcγ-receptors and C1q, eliminating ADCC/CDC while preserving neutralizing potency [75].

In 2024, an important addition to this list was the 17T2 monoclonal antibody. 17T2 is a broadly neutralizing monoclonal antibody targeting a conserved region of the spike receptor-binding domain (RBD). Isolated from a convalescent individual, 17T2 effectively neutralizes multiple variants, including Omicron sublineages, due to its broad RBD interaction. A structural analysis via cryo-EM revealed its binding mechanism, explaining its superior neutralization compared to similar mAbs. In vivo studies in K18-hACE2 mice demonstrated its prophylactic and therapeutic efficacy, significantly reducing viral loads and lung damage. The study highlighted 17T2’s potential for future clinical applications [76].

#### 4.2.3. Therapeutic Targets for Reactivated Reservoirs: The Case of EBV

In cases of reactivation from viral reservoirs, multiple therapeutic options are feasible depending on the literature and specific case. For Epstein–Barr virus (EBV) reactivation, available clinical experience supports nucleoside analogs such as ganciclovir and valganciclovir [77,78,79,80]. Nucleotide analogs are also effective [81,82,83], as well as treatments addressing potential co-activation with cytomegalovirus (CMV) [84]. Additional options include pyrophosphate analogs [85,86], protein kinase inhibitors [87,88,89], and EBNA1 inhibitors [90].

#### 4.2.4. Cardiovascular Regeneration

Endothelial progenitor cell (EPC)-based strategies have demonstrated significant promise in both preclinical and clinical settings. They improve myocardial function, reduce cardiac fibrosis, and accelerate wound healing through enhanced neovascularization and beneficial immune cell recruitment [91,92]. EPCs rapidly mobilize to ischemic myocardium, exerting cardioprotective effects via nitric oxide synthase activity [93], and support neovascularization in infarcted hearts [94]. Clinically, the intramyocardial injection of autologous CD34+ progenitor cells yielded favorable outcomes in refractory angina, confirming EPCs’ translational potential for vascular regeneration [95].

Nevertheless, several studies have flagged carcinogenic/tumorigenic concerns surrounding the clinical use of CD34^+^ endothelial progenitor cells (EPCs). Marçola and Rodrigues review how EPC-driven vasculogenesis can sustain tumor growth and metastasis [96], and [97] lists “malignant transformation” among the main safety risks of EPC transplantation. Emerging solutions are beginning to address these hazards. Precise CRISPR/Cas9 editing has already been achieved in primary human endothelial and EPC-like colony-forming cells [98] and has even been used to delete entire MHC loci in progenitor-derived endothelial cells to create safer, hypo-immunogenic grafts [99]. These advances in genome editing, together with parallel progress in ex vivo cellular reprogramming, suggest that trained or genetically safeguarded EPCs could soon be deployed while minimizing tumorigenic potential.

Additionally, combining human-induced pluripotent stem cell-derived endothelial cells (hiPSC-ECs) with mesenchymal stromal cells secreting SDF-1α has proven effective in regenerating ischemic cardiac tissue and improving heart function [100]. Moreover, mobilized bone marrow progenitor cells can deliver cytoprotective genes to damaged myocardium, enhancing functional recovery [101]. Recent advances in direct cellular reprogramming, such as the 7G-modRNA cocktail, have further expanded therapeutic avenues by inducing substantial cardiomyocyte-like cell formation, boosting cardiac performance and driving angiogenic mechanisms [102,103].

#### 4.2.5. Supplementation: Improvement Through the Use of PQQ, Ubiquinol, Resveratrol, and Others

Pyrroloquinoline quinone (PQQ) supplementation is recognized for its beneficial role in supporting mitochondrial function and cellular respiration, particularly in cases of hypoperfusion-induced damage, where both cellular and mitochondrial dysfunction are prominent. Its potential therapeutic role has been discussed as part of complementary supplementation strategies for managing deficits in cellular respiration within ischemic and hypoperfused tissues [104,105].

Ubiquinol is specifically noted for its positive effects on cellular respiration, particularly in conditions characterized by impaired mitochondrial energy production and cellular function. Its importance is emphasized within broader regenerative strategies aimed at counteracting cellular deficiencies caused by persistent ischemic or inflammatory conditions associated with hypoperfusion [106].

Resveratrol is widely recognized for its antioxidant properties, playing a crucial role in reducing reactive oxygen species (ROS) and mitigating oxidative stress-related damage. This is especially relevant in hypoperfusion conditions, where ROS accumulation and oxidative stress significantly contribute to tissue injury and impaired cellular function [107].

Lastly, taurine is a vital compound for maintaining mitochondrial health, cellular respiration, and antioxidant balance, while also playing a protective role against ROS, hypoperfusion, and ischemia-induced damage. Its ability to stabilize mitochondrial function and enhance vascular perfusion makes it a promising therapeutic agent in metabolic and cardiovascular diseases [108,109].

When considering the use of these supplements in hypoperfusion-related conditions, certain precautions should be taken to ensure safety and efficacy. Pyrroloquinoline quinone (PQQ), while generally well tolerated, may lead to oxidative imbalances if taken in excessive doses, making appropriate dosage control and periodic assessment of oxidative stress markers necessary. Similarly, ubiquinol supplementation requires caution, as a high intake can affect coagulation pathways and interact with anticoagulant medications [110,111,112]. Resveratrol, despite its antioxidant benefits, has estrogenic properties [113] and may interfere with platelet aggregation, warranting careful use in individuals on antithrombotic therapy or those with hormone-sensitive conditions [114]. Lastly, taurine, though beneficial for mitochondrial function and vascular perfusion, may cause hypotension due to its vasodilatory effects [115]. These considerations highlight the importance of careful monitoring and individualized assessment when integrating these supplements into therapeutic protocols.

## 5. Regulatory Status, Contraindications, and Safety Considerations

With the exception of short courses of licensed antivirals for acute COVID-19, none of the medicines included in the Vaxtherapy schedule carry marketing approval for the long-term treatment of syndromes attributed to the vaccine spike protein. Oral fibrinolytic enzymes (nattokinase, lumbrokinase, serrapeptase, bromelain), systemic fibrinolytics (alteplase, streptokinase), spike-neutralizing monoclonal antibodies (tixagevimab with cilgavimab, bebtelovimab) and regenerative cell products (mesenchymal stromal cells, exosome preparations) could therefore be given under compassionate-use agreements or an IND, after ethics approval and written informed consent. The consent form must state clearly that these agents are investigational, and that local access rules have been met.

Notwithstanding the investigational status outlined above, several constituents of the protocol are already authorized (albeit for *different* clinical settings) for precisely the physiological function they are expected to deliver here. Alteplase and streptokinase, for example, have full approval for acute ischemic stroke, ST elevation myocardial infarction and high-risk pulmonary embolism, confirming both their fibrinolytic potency and their risk mitigation framework when used to dissolve intravascular fibrin [116,117].

As of June 2025, no spike-neutralizing monoclonal antibody is licensed for COVID-19 post-vaccine syndromes. Nevertheless, there is a growing concern about this disease [49,118,119,120,121], so dedicated trials in post-vaccine cohorts are expected to begin soon. Early-phase trials (such as the sipavibart study now recruiting long-COVID cohorts) have yet to include vaccine-related cases, so their use in longvax patients is confined to physician-sponsored IND, or compassionate use programs that must satisfy FDA 21 CFR 312.310 (or EU CTR Art. 83) and obtain local IRB approval [122]. Consequently, monoclonals can be given only within registered studies or expanded-access frameworks, with mandatory pharmacovigilance reporting. Likewise, the spike-neutralizing antibody pairs casirivimab + imdevimab and tixagevimab + cilgavimab obtained marketing authorization or emergency use listing in 2020–2022 for treatment or prophylaxis of SARS-CoV-2 infection; although variant escape has curtailed routine use, the underlying regulatory dossier (pharmacodynamics, toxicology, CMC) remains valid for any future indication centered on specific spike vaccine variant neutralization, rather than whole-virus suppression [70,123,124].

By contrast, mesenchymal stromal cells and exosome preparations are still confined to early-phase trials, so their deployment in the regenerative phase will necessarily require formal human or expanded-access protocols. Finally, adjunctive oral enzymes such as nattokinase or bromelain carry “generally recognized as safe” (GRAS) status as dietary supplements, providing a legally viable (though non-therapeutic) route of access while controlled studies on their antifibrin activity progress.

## 6. Considering Alternative Mechanisms Beyond the Proposed Model as a Basis for Future Therapeutic Strategies

Before analyzing the specific case of COVID-19 vaccines, it is useful to recall the lessons from the adjacent condition, long COVID. A persistent question in that field has been whether a single root cause exists or whether several primary drivers can converge on a similar clinical picture. Until well into 2024, the literature still entertained four ostensibly independent mechanisms: (1) persistent viral reservoirs, (2) de novo auto-immunity, (3) tissue-centered injury/dysregulation, and (4) latent-virus reactivation. More recent work, however, has largely converged on a common denominator: viral persistence appears sufficient to ignite the downstream pathways responsible for the other phenotypes [49], from [8] as well as previous evidence [15,125,126]. Although intuition does not always reveal the true a etiology, the novel variable introduced by SARS-CoV-2 is, by definition, the virus (and therefore its spike protein), irrespective of the many interaction routes. The intermediate steps in the ensuing cascade are complex and heterogeneous, and additional pathways may yet be uncovered.

Now, in the case of longvax, the syndrome is triggered by the vaccine constituents themselves, whether an adenoviral vector, an mRNA platform, or a subunit formulation that delivers the spike protein. The common element across all platforms (and the point of overlap with many long-COVID manifestations) is the spike protein. Although vaccine-specific factors can also produce discrete adverse events independent of spike, such as, for instance, the interaction between PF4 and the adenoviral polyanion capsid [27], several studies show that spike particles alone, through downstream cascade effects, can account for the other explanatory pathways: that is, de novo autoimmunity [127,128], the reactivation of latent viruses [46,56,129,130], and direct endothelial or tissue injury [1,2,131]. These intermediary steps are unlikely to be the sole mechanism on which therapy should be based; other routes may exist, and acknowledging them remains an important limitation of the present study.

Regarding how the spike protein, beyond its hemodynamic effects, can drive viral reactivations, several studies show that spike induces systemic immunosuppression: once it circulates in exosomes after mRNA vaccination, it represses the IRF7–STAT1 axis and dampens type I interferon signaling [132]; it also promotes excess TGF-β, which blocks CD8^+^- and NK-cell cytotoxicity and permits EBV reactivation [129]. Clinical relevance is suggested by cases of disseminated herpes zoster after vaccines that contain only spike, where the authors propose the protein itself as the common trigger [130], and by skin biopsies that demonstrate vaccine-derived spike inside keratinocytes in the reactivated lesion [56]. A recent review summarizes this spike hypothesis, noting that the protein’s systemic distribution and tissue affinity can explain shared adverse reactions, including viral reactivation, even when vectors and excipients differ [46]. Thus, the combination of hypoperfusion, interferon shutdown, and TGF-β-dependent cytotoxic anergy, together with other factors such as spike-induced syncytial reservoirs [13], may offer a unified framework for understanding why herpesvirus reactivation serves as a bridge phenotype between long COVID and longvax.

Pointing out the convergence of autoimmune phenomena observed across the various COVID-19 vaccine platforms, it seems that the spike protein drives a pathogenic pivot. With mRNA vaccines (BNT162b2, mRNA-1273), it has been shown that human anti-spike antibodies recognize dozens of tissue auto-antigens through structural mimicry [133]. In post-mRNA myocarditis, the presence of free-circulating spike correlates with cardiac injury [131], and cardiology consensus statements link the protein to epitopes on α-myosin and troponin [134]. De novo Graves’ disease [135] and autoimmune hepatitis [136] have also been reported, both attributed to mimicry or tissue persistence of spike. Adenoviral-vector vaccines show a parallel pattern: the expression of spike has been associated with MOG-associated disease (MOGAD), presumably via the activation of B-cells directed against myelin [137], and with Guillain–Barré syndrome, where spike adhesion to gangliosides is thought to trigger antiganglioside antibodies [138]. For the protein subunit vaccine NVX-CoV2373 (composed essentially of purified spike plus adjuvant) pharmacovigilance has detected a disproportionate signal of myopericarditis, of magnitude comparable to that seen with mRNA vaccines, reinforcing the notion that the antigen alone suffices to spark autoimmunity [139]. Taken together, these data support that, irrespective of the vector or formulation, wherever spike is present, the same families of autoimmunity (cardiac, neurological, hepatic, thyroid) emerge, underscoring its central role in the genesis of vaccine-related immune-mediated events.

## 7. Limitations of the Model for Vaxtherapy Protocol

The hypotheses underpinning this model draw on the pathophysiological framework developed in earlier works from 2022 and 2024 [4,8], focusing primarily on the actions of the spike protein and persistent mRNA instructions. These factors, in turn, initiate the subsequent cascade of vascular and immunological damage, pathological permeability, syncytial formation, and hypoperfusion, as well as pathogen reactivations (ranging from expansion of pathogenic reservoirs to tumorigenesis/carcinogenesis). It is possible that additional, concurrently unaccounted factors could improve outcomes if they were incorporated into this pathophysiological model. Moreover, the appropriate selection and analysis of biomarker data may substantially impact the protocol’s evaluation, alongside other methodological considerations, such as study population size, the use of triple-blind randomized controlled trials, and other measures that influence the overall quality of the research.

## 8. Discussion

In summary, we have leveraged our understanding of the pathophysiology of long COVID (based on both our 2022 and 2024 studies as well as external sources) to accelerate the development of a strategy that may at least partially resolve longvax. This is particularly relevant for longvax characterized by spike persistence (LC-B, according to the 2022 model), yet it also holds sufficient theoretical potential to be applied to cases involving persistent mRNA instructions (LC-C2) even though that variant requires further research.

In any case, it requires a multipronged strategy grounded in each patient’s pathophysiological context. First, it is postulated that fibrinolytic therapy should be the first line of action that can mitigate the microvascular hypoperfusion perpetuated by fibrin amyloid-resistant microclots (FARMs). By restoring capillary flow and nutrient exchange, it establishes a prerequisite for subsequent interventions that otherwise would not reach the proper destination effectively. Second, neutralizing residual spike through multimodal monoclonal approaches (e.g., sotrovimab, casirivimab, imdevimab) is particularly relevant in LC-B (post-vaccine) cases, where mutational variability is comparatively limited. Neutralizing the spike not only curtails direct cytopathic effects but also alleviates excessive immune activation, creating a more favorable environment for pathogen clearance. Once the microclot burden and spike persistence have been addressed, targeted antimicrobial or antiviral treatments become more effective in controlling reactivated pathogens, thus preventing further immune system distraction or tissue damage. Finally, supplementation and regenerative approaches can promote the recovery of compromised cell populations and support tissue repair, complementing the foundational steps of fibrinolysis, spike neutralization, and pathogen clearance.

Naturally, the proposed protocol is grounded in both the pathophysiological understanding from the literature and the clinical experience with each proposed agent. This tiered approach underscores the significance of treatment sequence and timing; wrongly commencing supplementary, multi-monoclonal action (not to be confused with polyclonal action), or regenerative measures before resolving persistent microclots, spike-induced hypoperfusion, and damage could limit therapeutic efficacy or even facilitate the resistance of previous existing pathological reservoirs. Overall, the proposed protocol aims to restore functional homeostasis (metabolic, immune, and structural) while minimizing patients’ reliance on prolonged pharmacological interventions. Likewise, alternative mechanistic explanations and additional study limitations have been considered in brief. Further research, particularly randomized trials incorporating these multiphase interventions, will be crucial to refining best practices and achieving consistent clinical outcomes in both long COVID and longvax populations.

## Figures and Tables

**Figure 1 diseases-13-00204-f001:**
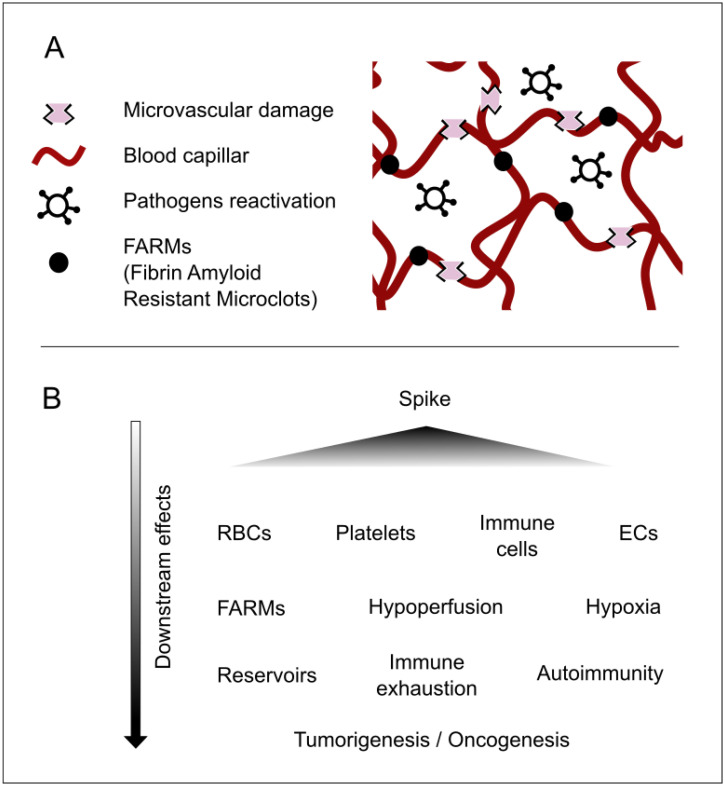
Major concepts underlying the pathophysiology of longvax for developing and advancing the Vaxtherapy approach. (**A**) Key microscale elements that reflect the progression of post-vaccine injury. The spike protein (or the persistence of its coding instructions) produces the already documented harm. These are responsible for the FARMs and for damage that is both direct, targeting vascular related cells such as endothelial cells (ECs), red blood cells (RBCs), platelets, and monocytes; and indirect, via immune exhaustion, syncytia formation, and inhibition of nutrients and defenses caused by hypoperfusion. This environment becomes the perfect breeding ground for reservoir strengthening and pathogenic reactivations. This process may contribute to additional injuries, ranging from neuroimmune and endocrine disruptions to intestinal permeability and nearly all damage observed in long COVID. (**B**) A cause-and-effect diagram illustrating the cascade of consequences in longvax, aiding in the design of the Vaxtherapy therapeutic strategy.

**Figure 2 diseases-13-00204-f002:**
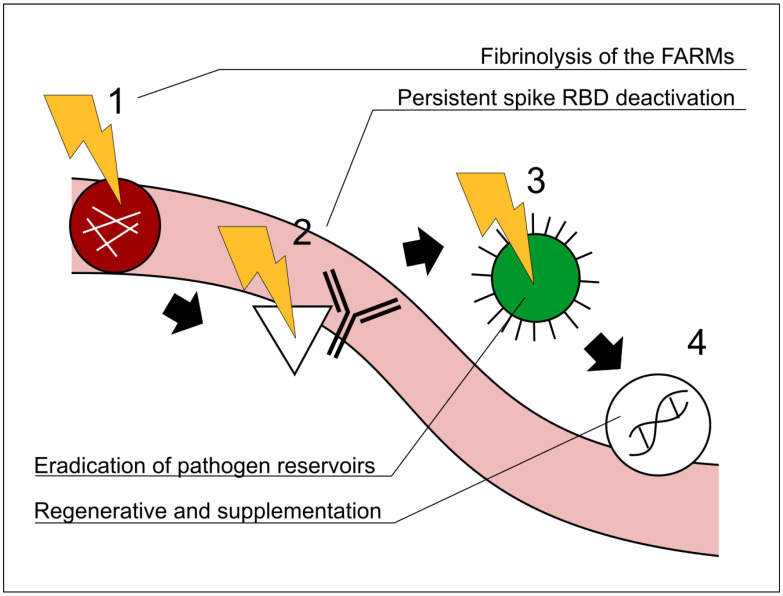
Representative schematic illustrating the recommended order of interventions in the Vaxtherapy protocol. A precedence relationship among the steps is shown by black arrows and the step numbers 1–4. The order in which the steps are applied is critical. Section 4 explains the relationship among the steps in greater detail. FARMs: fibrin amyloid-resistant microclots; RBD:receptor-binding domain. 1 Fibrinolytic therapy to restore immunological and therapeutic access to regions entrenched in hypoperfusion (e.g., reservoirs). 2 Monoclonal cocktail approach targeting the spike proteins, receptor-binding domain region. 3 Antiviral and broad antipathogenic measures directed at reservoir areas previously occluded by FARMs. 4 Regenerative and supplementary therapies, once the barriers addressed in steps 1, 2, and 3 have been removed. Note that steps 1–4 are not independent processes but follow a strategic order of precedence. For example, initiating step 3 (antiviral therapy) before completing steps 1–2 would leave micro-thrombotic obstructions in place, preventing the drug from effectively reaching the virus’s most protected niches; sub-therapeutic exposure could then foster antiviral resistance and aggravate the clinical course. Similarly, implementing step 4 while skipping step 3 would make the regenerative intervention largely futile, as successive reactivations would reproduce the very lesions that step 4 seeks to repair.

**Figure 3 diseases-13-00204-f003:**
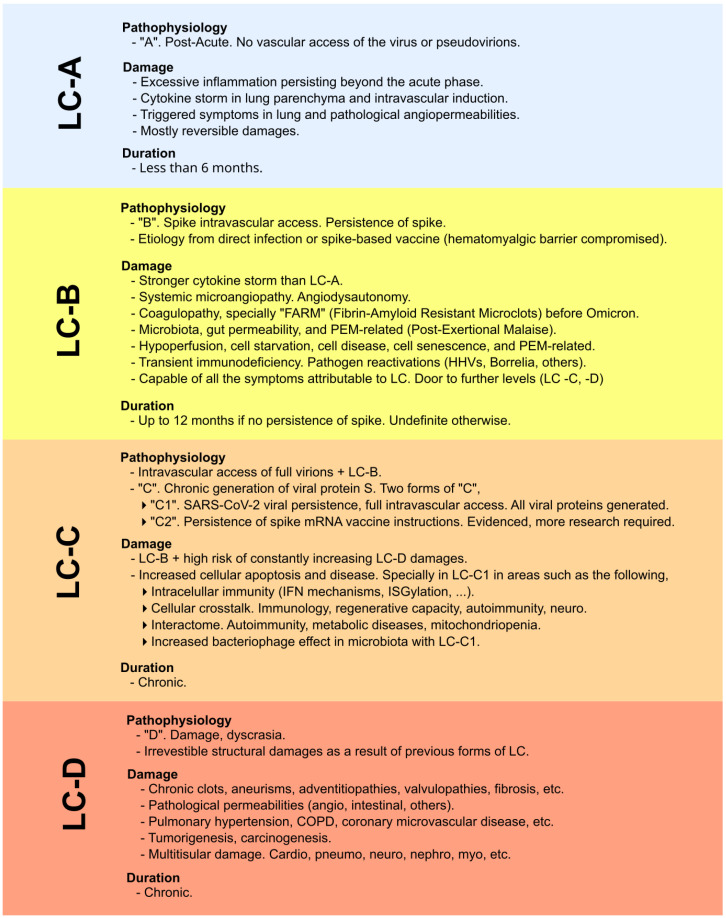
Classification and explanation of the different scenarios, focusing not on symptoms but on the pathophysiological context and impact in which viral proteins are involved (whether originating from infection or vaccination). Original classification from [8]. LC-A: no intravascular access of viral particles; LC-B: intravascular access and persistence of the spike protein; LC-C: chronic spike generation, either from SARS-CoV-2 persistence or from sustained expression driven by COVID-19 mRNA vaccine instructions; LC-D: irreversible structural damage caused by the preceding stages.

**Figure 4 diseases-13-00204-f004:**
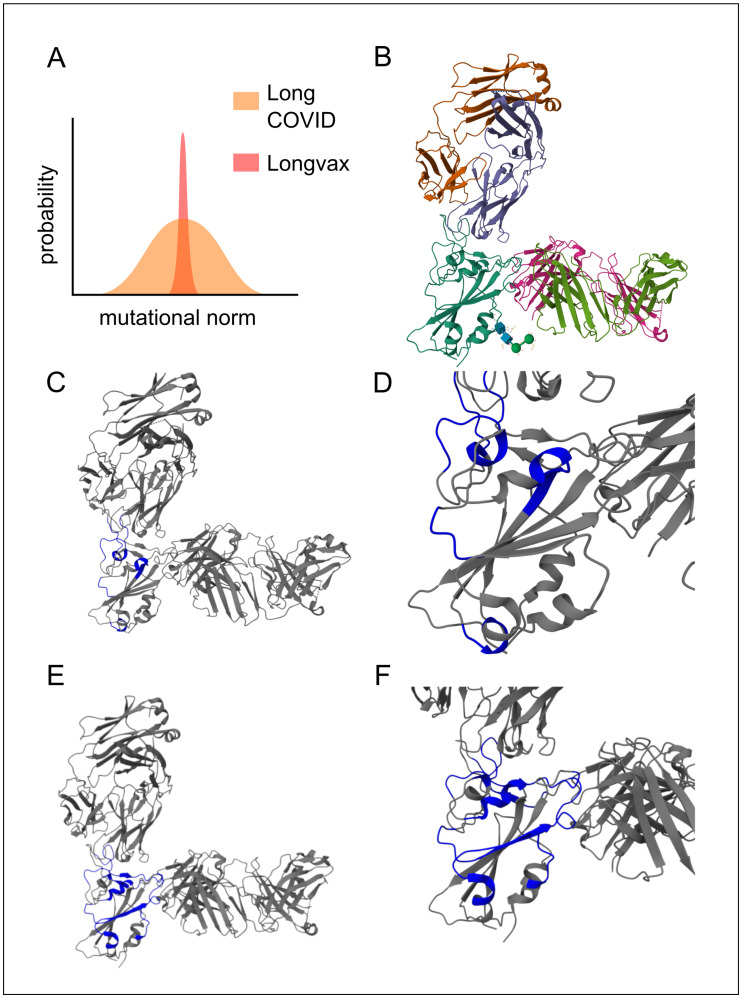
Multimodal monoclonal strategy for Vaxtherapy. Synergistic effect of co-administering casirivimab and imdevimab monoclonal antibodies against pre-Omicron SARS-CoV-2 spike variants in the region with the greatest functional stress and pathogenic impact: the receptor-binding domain (RBD), image from the RCSB PDB (RCSB.org) of PDB ID 6XDG [50]. (**A**) Conceptual image highlighting the theoretical mutational constraints of spike persistence in “longvax” compared to long COVID, which motivates the approach of the application of monoclonals such as casirivimab or imdevimab. (**B**) Overall view of the polyclonal conformation. Gold and purple represent the light and heavy chains of casirivimab, respectively; cyan shows the spike RBD; green and pink denote the light and heavy chains of imdevimab. (**C**) Highlighted in blue, the interactions of casirivimab with the spike RBD residues, 386–390, 399–406, 416–420, 426–430, 453–460, 468–474, and 486–489, overlapping the ACE2 (Angiotensin-Converting Enzyme 2) binding ridge. (**D**) Close-up of the docking interaction between the RBD and casirivimab. (**E**) Highlighted in blue, the interactions of imdevimab with the spike RBD residues: 369–386, 405–411, 416–417, 426–446, 449–460, and 480–500, in the portions on the outer flank of the RBD (receptor-binding domain). (**F**) Close-up of the docking interaction between the RBD and imdevimab.

## Data Availability

Data supporting the co-administration of casirivimab and imdevimab as SARS-CoV-2 monoclonals applied on mice. The corresponding structural data PDB ID 6XDG, mentioned in Figure 3, can be accessed via the RCSB PDB (RCSB.org) at https://doi.org/10.2210/pdb6XDG/pdb. This structure is originally described in: Hansen, J.; Baum, A.; Pascal, K.E.; Russo, V.; Giordano, S.; Wloga, E.; Fulton, B.O.; Yan, Y.; Koon, K.; Patel, K.; et al. *Science*
**2020**, *369*, 1010–1014. https://doi.org/10.1126/science.abd0827.

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
