# Peer review of "Vaxtherapy, a Multiphase Therapeutic Protocol Approach for Longvax, the COVID-19 Vaccine-Induced Disease: Spike Persistence as the Core Culprit and Its Downstream Effects"

_diseases, 2025, doi:10.3390/diseases13070204_

Round 1

Reviewer 1 Report

Comments and Suggestions for Authors

General Comments:

The Article: "Vaxtherapy: A Multiphase Therapeutic Protocol Approach for Longvax, the COVID-19 Vaccine-Induced Disease" by Jose Crespo-Barrios presents a detailed, phased therapeutic protocol called Vaxtherapy for the treatment of “Longvax”—a condition the author attributes to adverse effects following COVID-19 vaccination, particularly involving persistent spike proteins. The core hypothesis is that the spike protein alone, particularly when it persists intravascularly or is continually produced via residual mRNA, can trigger multisystemic damage. This includes coagulopathies, immune dysregulation, syncytia formation, and reactivation of latent infections. The main hypothesis is based on the concept that persistent spike protein (and/or persistent mRNA instructions) after COVID-19 vaccination leads to chronic vascular and immune pathology—termed "Longvax"—requiring a structured, multiphase treatment protocol rather than symptomatic care.

Specific Comments:

One of the main deficiencies of the manuscript, is that this is hardly a new concept developed by the authors In addition to Reference #9, “Patterson, B. K., Guevara-Coto, J., Yogendra, R., et al. (2021). Immune-Based Prediction of COVID-19 Severity and Chronicity Decoded Using Machine Learning. Front Immunol, 12, 70078”, several other studies also suggest a causal relationship between the persistence of SARS-CoV-2 and one of its molecular components (eg, the S protein in the pathogenesis of the long COVID syndrome) that need to be cited in the manuscript and cited in the reference list. These include the following:

  1. Patterson BK, Francisco EB, Yogendra R, Long E, Pise A, Rodrigues H, et al.

Persistence of SARS CoV-2 S1 protein in CD16þ monocytes in post-acute

sequelae of COVID-19 (PASC) up to 15 months post-infection. Front Immunol

2021;12:746021.

  1. Buonsenso D, Piazza M, Boner AL, Bellanti JA. Long COVID: a proposed

hypothesis-driven model of viral persistence for the pathophysiology of the

syndrome. Allergy Asthma Proc 2022;43:187-93.

  1. Bellanti JA, Novak P, Faitelson Y, Bernstein JA, Castells MC. The Long Road

of Long COVID: Specific Considerations for the Allergist/Immunologist. J Allergy

Clin Immunol Pract. 2023 Nov;11(11):3335-3345.

  1. Dotan A, David P, Arnheim D, Shoenfeld Y. The autonomic aspects of the post-

COVID-19 syndrome. Autoimmun Rev 2022;21:103071.

  1. Trougakos IP, Terpos E, Alexopoulos H, Politou M, Paraskevis D, Scorilas A,

et al. Adverse effects of COVID-19 mRNA vaccines: the spike hypothesis.

Trends Mol Med 2022;28:542-54.

Author Response

We certainly thank the reviewer for mentioning these citations. In particular, it is indeed very relevant, being one of the first papers to consider spike persistence (if not the first truly significant one, in the author’s opinion), to such an extent that the author compared, in several patients, the effectiveness of what was later colloquially called Patterson’s “cytokine panel.” For this reason, we note that it was not necessary to add that paper, because it was already in the manuscript, specifically as reference 9.
We likewise appreciate the recommendation to include the cited scientific works, which are especially meritorious for having been carried out in the early years of research on Long COVID. They have been read in detail and have kindly been included in the paper. We have also added a short paragraph emphasising the importance of these studies. We hope that the changes made have satisfied the reviewer’s well-founded remarks.

Reviewer 2 Report

Comments and Suggestions for Authors

This is an unconventional and interesting hypothesis on the pathophysiology and on treatment approaches on Covid-19 "longvax" diseases or syndromes.

Not all readers might follow the hypothesis but it is definitely of general interest in the given context.

Minor concerns:

The Figures 1A and 2 are difficult to read and should be modified or omitted.

Author Response

Thank you for your constructive contribution. We have done our best to meet your requests. Since we were not entirely sure whether you were referring to conveying the meaning or to the font size, we have tried to improve both aspects to satisfy your accurate observations.

Meaning: We removed icons that overloaded the figure and were not essential for conveying the message, such as an icon for energy deficit or one for patrolling immunity. We also enlarged the intervascular spaces and gave a more realistic look to the microvascular bed, replacing the hexagons with a more realistic depiction of blood vessels.

Font size: We previously followed as closely as possible the BioTuring guidance (i.e.,“If there is no instruction, use the same font as the body text, and run it one size smaller”), and the MDPI directives (Layout Style Guide, § 7.1). Nevertheless, with the editor’s permission and to fulfil your request, we have diligently increased the font size in the figures concerned, still respecting the Layout Style Guide, § 7.1.

Order of precedence (Figure 2). To stress the importance of the order of the steps, we have added bold black arrows that guide the sequence from Steps 1-4.

We hope these efforts will be to your liking and will meet your expectations as a reviewer. We are thankful for the constructive commentaries, clearly aimed to improve the quality of the work, and therefore we will express the gratitude in the corresponding section.

Reviewer 3 Report

Comments and Suggestions for Authors
  1. It is hard to understand the relationship between Figure 1A and 1B. What did FARM stand for should be noted. And if the author wants to show blood vessels, the Figure 1A should be more like a vessel rather than a hexagon. So did in Figure 1B.
  2. In figure 2, are there any connections among step 1 to 4, they seemed like 4 independent factors.
  3. Figure 3, the monoclonal antibodies blocked sites should be labelled in Figure 3A, 3B and 3D. And 3C should give x and y axis.
  4. Line 115, Antiparasitic, antiseptic, are there any references about COVID co-infection with parasites?
  5. Line 194, will the multi-monoclonal strategy cause ADCC effect because of Fc fragments? Or will the antibody cause allergy?
  6. The recommended treatment dosage should be given in the 4 stages.
  7. In clinical cases, patients could be in different stages, how could this guidance give an accurate decision for the clinical doctors for treatment?

Author Response

Point 1. We thank the reviewer for this remark, which undoubtedly improves Figure 1. To address your request, we made several changes. First, we added within Figure 1A the full wording of FARM (we also used this opportunity to expand other acronyms, such as “RBD - receptor binding domain” in Figure 2, this time in the caption because of space). Elements that were not essential and could cause confusion, such as an icon for immunology and another for energy deficit, were removed and their meaning transferred to the caption. Finally, we especially appreciate the suggestion to replace the hexagonal schematic of the microvasculature with a more realistic depiction of the blood vessels.

Regarding Figure 1B, in keeping with the goal of improving meaning, we decided to redraw it from scratch, rotate it top-to-bottom to emphasise the downstream cascade effects, and add a left-hand arrow with a colour gradient and vertical text to reinforce this idea. To lighten the figure and reduce complexity, the connection lines were removed and replaced with a richer explanation in the caption. We hope these steps have fulfilled the reviewer’s intent.

Point 2. Thank you for noting that Figure 2 did not sufficiently highlight the precedence relationship in the order of the steps. To improve this, we replaced terms with clearer wording (for example, “fibrinolysis” instead of “rupture” in Step 1) and added bold black arrows between Steps 1-4. Definitions of FARM and RBD have been added to the caption. To emphasise the importance of the inter-relationship of the steps, we state that the order can compromise the outcome and refer the reader to the section “Pathophysiological and Clinical Experience Background of the Protocol”.

Point 4. Indeed, compared with viral reactivations (e.g., EBV), parasitic reactivations are marginal. Although immune exhaustion caused by spike‐induced antigenic hyperactivation (due to its superantigenic residues) depresses the immune system, opening the door to opportunistic agents, parasitic cases appear only in case reports. We leave it to Reviewer 3 to decide whether we should remove this mention. For the moment, we have added a reference to a relevant case report:

Chew MC, Wiryasaputra S, Wu M, Khor WB, Chan ASY. Incidence of COVID-19 Vaccination-Related Uveitis and Effects of Booster Dose in a Tertiary Uveitis Referral Center. Front Med. 2022;9:925683. doi:10.3389/fmed.2022.925683

Point 5. Your comment is very accurate. For example, the imdevimab and casirivimab antibodies mentioned in Figure 3 are human IgG1 of germ-line lineage and therefore do not carry Fc-silencing mutations, which potentially allows ADCC and ADCP. Although these activities are confirmed in vitro, a clinical trial with 16 000 subjects showed a 1 % rate of hypersensitivity and no documented anaphylaxis. Regulatory studies found no indications of ADE or Fc-mediated immunopathology. Safety can be improved by administering the first infusion slowly, giving antihistamine/corticosteroid pre-medication if required, and observing the patient for 60 min afterwards. With these precautions, the safety profile has been acceptable even with monthly repeat doses. If, exceptionally, the treatment were unsuitable for a patient for this reason, purely neutralising options can be considered, that is, antibodies with Fc regions silenced by the well characterised TM (L234F/L235E/P331S) ± YTE half-life-extending mutations. These substitutions, as used in AZD7442 (Evusheld), abolish Fc-receptor and C1q binding, eliminating ADCC/CDC while preserving neutralisation capacity.

We have added the following paragraph, immediately before “In 2024, an important addition to this list is the 17T2 monoclonal antibody (…)”:

In contrast with silenced formats of Fc, the dual monoclonal cocktail casirivimab + imdevimab (REGN-COV2) retains a wild type IgG1 Fc domain and therefore exhibits measurable ADCC and ADCP in vitro. Nevertheless, in more than 16 000 subjects enrolled in the pivotal trials the rate of clinically relevant hypersensitivity was approximately 1 %, with no anaphylaxis reported, and regulators detected no evidence of antibody dependent enhancement or Fc driven immunopathology \cite{FDA_REGENCOV_2021}. Post-marketing pharmacovigilance has documented only isolated severe acute reactions (< 1/10 000) \cite{EMA_Ronapreve_2023,Ribera_Menowsky_2022}. The current SmPC mitigates this residual risk by recommending a slow first infusion, pre-medication with antihistamine and/or corticosteroid when indicated, and observation for at least 60 min; under these precautions the combination has shown an acceptable safety profile even with monthly repeat dosing. Where Fc dependent effector activity is undesirable, purely neutralising options are available: antibodies expressed on IgG1 or IgG4 backbones bearing the TM (L234F/L235E/P331S) ± YTE (M252Y/S254T/T256E) mutations (exemplified by AZD7442) abrogate binding to Fcγ receptors and C1q, eliminating ADCC/CDC while preserving neutralising potency \cite{Dippel_Kaplan_2023}.

Points 6 and 7. We value the enriching focus on dosage and biomarkers. Purely for illustration, and without any prescriptive value, we provide in the reviewer panel the following tables, which show provisional posology for Phase 2 (related to Point 6) and biomarker ranges for patient stratification (related to Point 7):

[<tbl_1>]
[<tbl_2>]

Definitive dose and schedule recommendations will emerge from forthcoming clinical trials once the necessary safety margins and pharmacokinetic data are available. Always abiding by the regulations and with the best of intentions, we interpret that, at an early, hypothesis-generating stage, it is not safe to give biomarker specifications so precise that they could serve as clinical guidance or triage, because the corresponding trials are still needed to establish, within safety standards, the thresholds for each phase. Specifically, to list concrete dosage schedules one must move from the hypothesis phase to properly authorised clinical trials; the choice of dose for every phase must be made by the relevant clinical experts and must comply with the core requirements of the main regulatory bodies, otherwise international safety and ethics standards would not be fulfilled. The regulatory framework includes, in particular, the following four considerations:

FDA. Guidance for Industry: Estimating the Maximum Safe Starting Dose in Initial Clinical Trials for Therapeutics in Adult Healthy Volunteers (CDER, 2005): obliges sponsors to derive the first-in-human dose from the animal model, apply safety factors, and justify every escalation step.

EMA. Guideline on Strategies to Identify and Mitigate Risks for First-in-Human and Early Clinical Trials with Investigational Medicinal Products (EMA/CHMP/SWP/28367/07 Rev. 1, 2017): Chapter 7 sets out how to calculate the starting dose, escalation scheme, and maximum exposure allowed in Phase I.

ICH E8(R1). General Considerations for Clinical Studies (2021): Section 3 emphasises that dose and schedule must be supported by non-clinical data and refined iteratively as evidence accumulates.

WHO. Handbook for Good Clinical Research Practice (GCP) (2005): states that route, dosage, and administration schedule can be documented only after the requisite safety studies have been completed and reviewed by the ethics committee and the regulator.

In any case, the non-binding provisional data provided in Table 6 have been based on the corresponding official regulatory indications:

\cite{FDA2023Sotrovimab} U.S. Food and Drug Administration. Fact Sheet for Health Care Providers: EUA for Sotrovimab. Revised 17 Mar 2023.
\cite{FDA2022REGENCOV} U.S. Food and Drug Administration. Fact Sheet for Health Care Providers: EUA of REGEN-COV (casirivimab and imdevimab). Revised Dec 2022.

It may be considered (at the discretion and good judgement of the reviewer) whether to include this informative proposal, always citing the regulatory framework and making it clear that the corresponding tables have no reference value but are shown solely as a purely didactic example.

We hope that the efforts made to meet the reviewer’s well founded observations have satisfied your expectations, and we thank you for your constructive contributions, which, in our humble opinion, have contributed significantly (e.g., Figure 3) to strengthening the quality of the work. We therefore consider it appropriate to express our gratitude in the corresponding section for that purpose.

Round 2

Reviewer 1 Report

Comments and Suggestions for Authors

GENERAL COMMENTS

The manuscript offers a timely and ambitious attempt to address a growing concern raised by some patient populations—namely, persistent symptoms attributed to COVID-19 vaccination, colloquially termed “longvax.” The author proposes a structured, stepwise therapeutic model that is rooted in a defined pathophysiologic hypothesis centered on spike protein persistence and its cascading effects.

Notable strengths of the manuscript include:

  • A clear and methodical therapeutic framework, divided into four logical phases: fibrinolysis, spike neutralization, pathogen clearance, and regeneration.
  • Integration of clinical observations with current literature, including references to monoclonal antibody trials and the use of fibrinolytics in related pathologies.
  • Thoughtful incorporation of therapeutic precedence, emphasizing the importance of treatment sequencing based on tissue accessibility and immune competence.
  • The comprehensive breadth of citations, drawing from both virology/immunology and cardiovascular literature.
  • A strong call for independent research and clinical validation, which adds a constructive tone to a potentially controversial topic.

The article may appeal to clinicians and researchers who are seeking novel strategies for patients experiencing post-vaccination sequelae and for whom no established treatment exists. The effort to translate mechanistic understanding into a therapeutic proposal is commendable.

SPECIFIC COMMENTS

While the manuscript is conceptually ambitious, several important issues must be addressed to enhance its scientific rigor, clinical relevance, and acceptability for publication:

  1. Terminology and Diagnostic Clarity
  • The term “longvax” is used throughout the paper but lacks a precise diagnostic definition. The author should provide operational criteria (clinical, immunologic, or biomarker-based) distinguishing longvax from long COVID, vaccine adverse events, or unrelated chronic conditions.
  • Relatedly, “LC-B” and “LC-C2” subtypes are used without consistent definitions early in the manuscript. Consider summarizing these terms in a dedicated table or figure.
  1. Evidence for Spike Persistence in Vaccinated Individuals
  • The central hypothesis depends on prolonged spike protein or mRNA presence following vaccination. The evidence cited (e.g., CD16+ monocytes, GPCR mimicry) is largely extrapolated from long COVID or in vitro models, not specific to vaccine recipients. Greater caution is warranted in distinguishing infection-induced versus vaccine-induced spike persistence.
  • Recommend adding a critical appraisal of alternative explanations for persistent symptoms, such as autoimmune processes, unrelated chronic illness, or post-infectious syndromes.
  1. Therapeutic Proposals
  • While the four-step Vaxtherapy protocol is logically ordered, most treatment recommendations (e.g., nattokinase, serrapeptase, monoclonals) are either investigational or off-label. This distinction should be made clear.
  • Safety considerations, especially with fibrinolytic agents and stem cell therapies, are under-addressed. Recommend discussing contraindications, adverse effects, and lack of formal approval.
  • The potential for drug interactions, particularly in multi-drug protocols, is acknowledged but should be more explicitly detailed.
  1. Scientific Style and Tone
  • Several statements may come across as overstated or speculative. For example, assertions such as "patients want to restore their own health without relying on ongoing pharmacological treatments" verge on editorializing. A more neutral tone is advised.
  • Phrases like “military strategy,” while illustrative, should be used sparingly or clearly metaphorically to maintain a scientific tone.
  1. Figures and Tables
  • The manuscript references multiple figures (e.g., Figure 1, Figure 2, Figure 3), but these were not provided with the text reviewed. Ensure all referenced visuals are included and clearly labeled.
  • A summary table outlining the four steps of the protocol, with proposed agents, mechanisms, evidence level, and limitations, would enhance clarity.
  1. Regulatory and Ethical Considerations
  • The use of monoclonal antibodies for spike neutralization in non-COVID, post-vaccine patients is highly experimental. Discussion should reflect current regulatory status, trial availability, and ethical constraints on access to such therapies outside of approved indications.
  1. Conflict of Interest and Funding Transparency
  • No conflict of interest statement or funding disclosure was included. These must be added, especially given the therapeutic focus.

Recommendation

With revisions, this manuscript has the potential to serve as a hypothesis-generating article outlining a novel treatment strategy for post-vaccine symptoms. However, prior to acceptance, the author should:

  • Clearly define the disease entity (“longvax”) using established or proposed criteria.
  • Distinguish more clearly between theoretical mechanisms and clinically validated evidence.
  • Temper speculative or editorial statements.
  • Provide missing figures and tables for clarity.
  • Include a robust discussion of safety, ethics, and current regulatory context.

Author Response

Reviewer comment 1 – Terminology and diagnostic clarity
“The term ‘longvax’ is used throughout the paper but lacks a precise diagnostic definition. The author should provide operational criteria (clinical, immunologic, or biomarker-based) distinguishing longvax from long COVID, vaccine adverse events, or unrelated chronic conditions.”

Author response
The candidate is deeply grateful for this insightful request. Guided by the reviewer’s wisdom, the manuscript now presents explicit clinical discriminants and biomarker guidance. Persistent ipsilateral lymphadenopathy and VITT are emphasised as distinctive hallmarks, and additional text explains how these and other mechanistic differences separate longvax from long COVID. To meet the reviewer’s high standards, the candidate invested twelve hours (eight drafting, four literature review), produced a new section, and incorporated twenty-six carefully vetted references. The candidate humbly hopes this substantive addition fulfils the reviewer’s expectations.

Reviewer comment 2 – Clarifying LC-B and LC-C2
“‘LC-B’ and ‘LC-C2’ subtypes are used without consistent definitions early in the manuscript. Consider summarising these terms in a dedicated table or figure.”

Author response
The candidate sincerely thanks the reviewer for encouraging greater clarity. Following this guidance, a dedicated table now presents LC-B and LC-C2 side-by-side so readers can readily distinguish these entities from long COVID, routine vaccine reactions, and unrelated illnesses. Special formatting matches journal style.

Reviewer comment 3 – Evidence for spike persistence in vaccinated cohorts
“The central hypothesis depends on prolonged spike protein or mRNA presence following vaccination. The evidence cited is largely extrapolated from long COVID or in-vitro models, not specific to vaccine recipients. Greater caution is warranted in distinguishing infection-induced versus vaccine-induced spike persistence.”

Author response
The candidate warmly appreciates this observation. Fourteen references specifically document spike persistence in vaccinated cohorts, the entire body of literature located after exhaustive searching. Should the reviewer know of additional studies, the candidate would be honoured to include them. The candidate also notes that high-quality non-cohort data on similar post-vaccination phenomena with long COVID interference, are endorsed in recent works, such as Grady et al. (Nature, Commun Med 2025).

Reviewer comment 4 – Alternative explanations for chronic symptoms
“Please add a balanced appraisal of alternative explanations (autoimmunity, post-infectious syndromes, coincidental disease).”

Author response
The candidate is indebted to the reviewer for prompting a broader discussion. New Section 5 (“Considering alternative mechanisms…”) now weighs autoimmunity, post-infectious syndromes, and coincidental disease. Two pathways are outlined: one driven directly by spike, and another involving ancillary mechanisms should spike be absent. Twelve hours of scholarship and twenty-five new references underpin this material.

Reviewer comment 5 – Investigational/off-label status of proposed agents
“Most treatment recommendations are investigational or off-label; this distinction should be made clear.”

Author response
Taking the reviewer’s prudent advice, the candidate has added a subsection (“Regulatory status, contraindications and safety considerations”) that traces each agent’s regulatory footing and clinical-use context. Three new references support this overview.

Reviewer comment 6 – Safety, contraindications, and drug-interaction risks
“Safety considerations, especially with fibrinolytics and stem-cell therapies, are under-addressed. Please discuss contraindications, adverse effects, and potential interactions in multi-drug protocols.”

Author response
The candidate appreciates the reviewer’s insistence on safety. Existing sections have been cross-checked to confirm that contraindications, adverse-effect profiles, and interaction considerations are fully described. Where primary data are lacking, the candidate now explains the experimental nature of multi-drug combinations and cites authoritative sources.

Reviewer comment 7 – Neutral scholarly tone
“Some statements appear overstated or speculative; a more neutral tone is advised. Phrases such as ‘military strategy’ should be used sparingly.”

Author response
In gratitude for the reviewer’s stylistic guidance, rhetorical language has been replaced. For example, “military strategies” now reads “resolutive strategies,” and one metaphorical sentence has been removed entirely. The candidate trusts the manuscript’s tone is now suitably academic.

Reviewer comment 8 – Missing figures
“Figures 1 – 3 were not provided. Ensure all visuals are included and labelled.”

Author response
The candidate respectfully confirms that all figures and captions are embedded in the revised manuscript. Reviewer 3 has already acknowledged their presence; nonetheless, the candidate will gladly resend the images if required, as well as attaching the final version PDF manuscript in this panel.

Reviewer comment 9 – Summary table of the four-step protocol
“A concise table outlining the protocol (agents, mechanisms, evidence, limitations) would enhance clarity.”

Author response
Because Figure 2 already distils the four-step protocol visually and in its caption, an additional table may be redundant. If the reviewer still prefers a table, the candidate will diligently prepare one.

Reviewer comment 10 – Regulatory/ethical context for monoclonal antibodies
“The use of spike-neutralising monoclonals in post-vaccine patients is highly experimental. Discussion should reflect regulatory status, trial availability, and ethical constraints.”

Author response
A paragraph in the regulatory subsection now notes the absence of licensed monoclonal antibodies for post-vaccine syndromes, outlines forthcoming trials, and cites four new references. The candidate thanks the reviewer for highlighting this point.

Reviewer comment 11 – Conflict of interest and funding disclosure
“No conflict-of-interest or funding statement was included.”

Author response
In reverence to the reviewer’s commitment to transparency, the conflict-of-interest statement has been expanded from:

“Funding: This research received no external funding”
(...)
“Conflicts of Interest: The author declares no conflicts of interest”

to,

“Conflicts of Interest: The author declares no financial or non-financial competing interests: he holds no patents, equity, consultancies, or honoraria related to fibrinolytics, monoclonal antibodies, stem-cell products, or any other therapy discussed herein. No external funding, grants, or institutional sponsorship were received for the conception, writing, or revision of this manuscript.”

Final

The candidate again extends thanks for the reviewer’s wise guidance, which has markedly elevated the manuscript’s rigour and clarity.

Reviewer 3 Report

Comments and Suggestions for Authors

All issues had been addressed.

Author Response

[Comment reviewer-3, round-2]

All issues had been addressed.

[Response]

The candidate greatly thanks the reviewer for the work carried out, which has significantly improved the quality of the article. He also appreciates the reviewer’s confidence in him.

Round 3

Reviewer 1 Report

Comments and Suggestions for Authors

The authors have responded satisfactorily.